# Short-Term High-Starch, Low-Protein Diet Induces Reversible Increase in β-cell Mass Independent of Body Weight Gain in Mice

**DOI:** 10.3390/nu11051045

**Published:** 2019-05-10

**Authors:** Atsushi Masuda, Yusuke Seino, Masatoshi Murase, Shihomi Hidaka, Megumi Shibata, Takeshi Takayanagi, Yoshihisa Sugimura, Yoshitaka Hayashi, Atsushi Suzuki

**Affiliations:** 1Department of Endocrinology and Metabolism, Fujita Health University, Graduate School of Medicine, Toyoake 470-1192, Japan; a-masuda@fujita-hu.ac.jp (A.M.); sakai220@fujita-hu.ac.jp (S.H.); megumi03@fujita-hu.ac.jp (M.S.); haratake@fujita-hu.ac.jp (T.T.); sugiyosi@fujita-hu.ac.jp (Y.S.); aslapin@fujita-hu.ac.jp (A.S.); 2Departments of Endocrinology and Diabetes, Toyota Memorial Hospital, Toyota 471-8513, Japan; murase-m@med.nagoya-u.ac.jp; 3Department of Endocrinology, Research Institute of Environmental Medicine, Nagoya University, Nagoya 467-8601 Japan; hayashiy@riem.nagoya-u.ac.jp

**Keywords:** carbohydrate, β-cell mass, islet number, insulin, dietary intervention

## Abstract

Long-term exposure to a high starch, low-protein diet (HSTD) induces body weight gain and hyperinsulinemia concomitantly with an increase in β-cell mass (BCM) and pancreatic islets number in mice; however, the effect of short-term exposure to HSTD on BCM and islet number has not been elucidated. In the present study, we investigated changes in body weight, plasma insulin levels, BCM and islet number in mice fed HSTD for 5 weeks followed by normal chow (NC) for 2 weeks. BCM and islet number were increased in mice fed HSTD for 5 weeks compared with those in mice fed NC. On the other hand, mice fed HSTD for 5 weeks followed by NC for 2 weeks (SN) showed decreased BCM and insulin levels, compared to mice fed HSTD for 7 weeks, and no significant differences in these parameters were observed between SN and the control NC at 7 weeks. No significant difference in body weight was observed among HSTD, NC and SN fed groups. These results suggest that a high-starch diet induces an increase in BCM in a manner independent of body weight gain, and that 2 weeks of NC feeding is sufficient for the reversal of the morphological changes induced in islets by HSTD feeding.

## 1. Introduction

Insulin is secreted from pancreatic β-cells, and plays an important role in glucose metabolism. Glucose stimulates insulin secretion directly and indirectly through incretins such as glucose-dependent insulinotropic polypeptide (GIP) and glucagon-like peptide-1 (GLP-1) [1]. Both insulin and GIP contribute to the promotion of fat deposition in adipose tissue [2,3,4]. Indeed, any blockade of the GIP signal and/or hyperinsulinemia is a protection from high-fat diet (HFD)-induced obesity in mice [2,3,4,5,6]. On the other hand, we previously reported that both wild-type (WT) and GIP receptor-deficient (*Gipr*KO) mice fed a high-starch, low-protein diet (HSTD), show a body weight gain, compared with those fed normal chow (NC), indicating that HSTD induces obesity in a manner independent of the GIP signal [5]. In addition, glucose-induced insulin secretion (GIIS) from isolated islets is enhanced in WT and *Gipr*KO-fed HSTD, compared with those in WT and *Gipr*KO-fed NC or a moderate-fat diet 22 weeks after intervention of the diets [5]. 

These results suggest that the enhanced insulin secretion participates in body weight gain in mice fed HSTD. The increase of pancreatic β-cell mass (BCM) and islet number contributes to the augmentation of insulin secretion. Indeed, BCM and islet number are increased in WT mice fed HSTD, compared with those in WT mice fed NC 22 weeks after intervention of the diets [7]. It is reported that BCM is increased in obese and/or insulin-resistant rodents and humans [8,9]. Therefore, it is unclear whether BCM and islet number in WT mice fed HSTD for 22 weeks are increased, compared with those in mice fed NC due to excess supplementation of glucose, the final product of starch, or to obesity and decrease in insulin sensitivity. 

In the present study, we find that BCM and islet number are increased in WT mice fed HSTD for 5 weeks, compared with those fed NC, before the difference in body weight is apparent, and that the increase of BCM and islet number is reversible 2 weeks after switching from HSTD to NC, regardless of the change of body weight.

## 2. Materials and Methods 

### 2.1. Animals and Diets

Eight-week-old male C57BL/6J wild-type (WT) mice were obtained from CLEA Japan (Osaka, Japan) and housed in a room under a standard 12:12-h light/dark cycle with free access to food and water. WT mice were divided into two groups: Mice fed normal chow (NC) (starch 58%, protein 29%, soy oil 13% of total energy; CLEA Japan, Osaka, Japan) and high starch, low-protein diet (HSTD) (starch 74%, protein 13%, soy oil 13% of total energy; CLEA Japan, Osaka, Japan), as previously reported [5,7,10,11]. CE-2 (CLEA, Japan, Osaka, Japan) was used for NC (carbohydrate 58%, protein 29%, fat consisting of soy oil 13% of total energy, the carbohydrate consisting of starch). HSTD (starch 74%, protein 13%, soy oil 13% of total energy) was produced by adding 51 g CE-2 to 46 g starch and 3.1 g soy oil. NC diet was 3.87 kcal/g, and HSTD diet was 3.43 kcal/g. Mice were fed for 3, 5 or 7 weeks, and then analyzed independently. Data on 5 weeks of intervention were also obtained from mice in which islets were isolated. All procedures were carried out according to a protocol approved by the Institutional Animal Care and Use Committee of Fujita Health University.

### 2.2. Intraperitoneal Glucose Tolerance Test (IPGTT) and Insulin Tolerance Test (ITT)

IPGTT was performed 5 or 7 weeks after intervention of the diets. After 16 h of food deprivation, glucose (2 g/kg) was administrated to mice intraperitoneally. ITT was performed 5 or 7 weeks after intervention of diets. After 6 h of food deprivation, insulin (0.75 U/kg) was injected intraperitoneally to mice. None of the mice received both IPGTT and ITT.

### 2.3. Plasma Biochemical Analyses

Blood was collected from the tip of the tail. Blood glucose levels were measured with Antsense Duo (Horiba, Kyoto, Japan). Blood samples were centrifuged (13,500 rpm, 10 min, 4 °C) twice, and the collected plasma samples were stored at −80 °C until analysis. Plasma insulin levels were measured using a Mouse Insulin ELISA Kit (Morinaga Institute of Biological Science, Kanagawa, Japan) as previously reported [7]. 

### 2.4. Isolation of Ribonucleic Acid (RNA) and Quantitative Real-Time Reverse Transcription Polymerase Chain Reaction (RT-PCR)

Pancreata were perfused with collagenase 4.5 mg (cat.112490020001; Roche, Basel, Switzerland) in 3 mL RPMI medium (1×)1640 (11875-093; gibcl, Tokyo, Japan). The pancreata were removed and collected in tubes containing 5 mL HBSS-HEPES(pH 7.2), and placed in a water bath (37 °C) for 11.5 min. Samples with KRB (pH7.4 0.1%BSA) were then centrifuged (1000 rpm, 2 min, 4 °C) twice. The sediment was blended with 8 mL Histopaque 1119 (11191; SIGMA, St. Louis, MO, USA) and added to a 4 mL mixture of 2 mL Histopaque 1119 and 2 mL Histopaque 1077 (10771; SIGMA, St. Louis, MO, USA) together with 4mL KRB (pH7.4 0.1%BSA). 

The adulterant was then centrifuged (1000 rpm, 25 min, 4 °C), the supernatant was collected and centrifuged again (1000 rpm, 2 min, 4 °C), and the islets were picked up. Total RNA was collected from isolated islets using the RNeasy Plus Kit (Qiagen, Tokyo, Japan) and complementary Deoxyribonucleic Acid (cDNA) synthesis was performed, as previously reported [5,7,12]. The polymerase chain reaction (PCR) method for thermal cycling was used with an Applied Biosystems by the 2^∆∆CT^ method. Each PCR reaction mixture (25 μL) contained 100 ng genomic DNA, SYBR qPCR Mix 12.5uL (Toyobo, Saitama, Japan), 3 μM of each primer 2.5 uL, Rob Reference Dye 0.5 uL (Toyobo, Saitama, Japan), and DEPC water 4.5 uL. The qPCR was performed with an initial polymerase activation step at 95 °C for 1 min, followed by 40 cycles of denaturation (95 °C for 15 s) and annealing/extension (60 °C for 45 s). The primer sequence is shown in Table 1. The expression levels of mRNA were normalized by those of *β-actin*.

### 2.5. Immunohistochemistry and Morphological Analyses

The pancreata of WT mice were fixed with 4% paraformaldehyde and embedded in paraffin. Serial sections of 4 μm thickness were cut from each paraffin block at 200 μm intervals and deparaffinized as previously reported [7,12]. The sections were incubated overnight at 4 °C with primary antibodies against insulin (1:300; ab7842; Abcam, Cambridge, MA, USA), followed by 90 minutes incubation in Alexa Fluor-conjugated secondary antibody (1:500; A11074; Alexa Fluor 546; Invitrogen, Grand Island, NY, USA) at room temperature. DAPI solution (1:2000; 340-07971; Dojindo, Kumamoto, Japan) was used at room temperature for 30 min. The total area of insulin-positive cells (β-cells) and the number of islets were determined with BZ-9000 fluorescent microscope system (Keyence, Osaka, Japan), as previously reported [7].

### 2.6. Statistical Analysis

Data are expressed as means ± SEM. Statistical significance of the difference between two groups was evaluated by Student’s *t*-test, and that for multiple comparisons was investigated by ANOVA using GraphPad Prism 7 for Windows (GraphPadSoftware, La Jolla, CA, USA) [7]. *p* < 0.05 was considered statistically significant.

## 3. Results

### 3.1. Pancreatic Islet Number and BCM Were not Increased in HSTD-Fed Mice 3 Weeks after Intervention

We first examined body weight, blood glucose levels and plasma insulin levels in WT mice fed NC or HSTD 3 weeks after intervention of diet. Body weight, blood glucose levels and plasma insulin levels under ad libitum-fed condition were not different between NC-fed mice and HSTD-fed mice (Figure 1A–C). Neither islet number nor BCM was different between NC-fed mice and HSTD-fed mice (Figure 1D–F).

### 3.2. Plasma Insulin Levels Were Elevated in HSTD-Fed Mice 5 Weeks after Intervention

We next investigated body weight, blood glucose levels and plasma insulin levels in WT mice fed NC or HSTD 5 weeks after intervention of diet. Body weight and blood glucose levels under ad libitum-fed condition were not different between NC-fed mice and HSTD-fed mice, but plasma insulin levels in HSTD-fed mice under ad libitum-fed condition were significantly higher compared with those in NC-fed mice (Figure 2A–C). To evaluate the insulin secretory response to glucose, IPGTT was performed 5 weeks after intervention of diet. Blood glucose levels and plasma insulin levels during IPGTT were not different between mice fed NC and HSTD (Figure 2D,E). We then conducted ITT to estimate insulin sensitivity in NC-fed and HSTD-fed mice 5 weeks after intervention of diet. Blood glucose levels in comparison to 0 min were significantly lower in mice fed HSTD, compared with those in mice fed NC (Figure 2F). 

### 3.3. Pancreatic Islet Number and BCM Were Increased in HSTD-Fed Mice 5 Weeks after Intervention

To assess the effect of a high-starch diet on pancreatic islets, we compared the morphology of islets in mice fed NC and HSTD 5 weeks after intervention of diet. Both islet number and BCM were significantly increased in mice fed HSTD, compared with those in mice fed NC (Figure 3A–C). 

We then analyzed the gene expression levels in islets of NC-fed and HSTD-fed mice 5 weeks after intervention of diet. The expression levels of *cyclin A2* and insulin receptor substrate-2 (*Irs2*) mRNA were significantly higher in the islets of HSTD-fed mice, compared with those in the islets of NC-fed mice (Figure 3D). 

### 3.4. Switching from HSTD to NC Decreased Pancreatic Islet Number and BCM

Mice were divided into three groups (Figure 4A). (1) NC group: Mice fed NC for 7 weeks (2) HSTD group: Mice fed HSTD for 7 weeks and (3) SN group: Mice fed HSTD for 5 weeks, and then fed NC for 2 weeks. We examined body weight, blood glucose levels and plasma insulin levels under ad libitum-fed condition 7 weeks after intervention of diet. Body weight and blood glucose levels were not different among the three groups (Figure 4B,C). Plasma insulin levels were significantly higher in HSTD group, compared with those in NC group and SN group, and were not different between NC group and SN group (Figure 4D). IPGTT or ITT was performed 7 weeks after intervention of diet. Blood glucose levels were not different among the three groups during IPGTT (Figure 4E). 

No significant difference in blood glucose levels in comparison to 0 min during ITT was observed among the three groups, except for the levels 60 min after insulin administration. Blood glucose levels in comparison to 0 min in SN and HSTD groups were significantly lower, compared to the NC group at 60 min after insulin administration (Figure 4F). We then investigated whether switching from HSTD to NC affected the morphology of islets independent of body weight and blood glucose levels. Both islet number and BCM in HSTD group were significantly increased, compared with those in NC and SN group, and were not different between NC and SN group (Figure 4G–I).

## 4. Discussion

In the present study, we show that BCM and islet number in mice fed HSTD for 5 weeks was increased, compared with that in mice fed NC, before the difference of blood glucose levels and body weight appeared, and that switching from HSTD to NC for two weeks reduced BCM and islet number, independent of any body weight change. Although plasma insulin levels in mice fed HSTD were significantly increased, compared to those in mice fed NC under ad libitum condition, insulin sensitivity evaluated by ITT was increased in HSTD-fed mice, compared to that in NC-fed mice. 

Continuous intravenous infusion of glucose or fat increases BCM in rodents [13,14,15,16]. Although there are many reports that analyze BCM in mice fed a high-fat diet (HFD), the morphology of islets in mice fed the high-starch diet has not been elucidated. Overnutrition, such as in HFD, induces obesity, together with a decrease of insulin sensitivity. Under these conditions, insulin demand is increased to sustain glucose homeostasis, and BCM is increased [17,18,19]. The morphology of islets and glucose metabolism in mice exposed to short-term HFD has been well investigated, as obesity and insulin resistance affects BCM, and it is not known whether excessive fat intake itself or obesity and/or insulin resistance increases BCM in mice under long-term exposure to HFD. 

In mice fed HFD, body weight, blood glucose and plasma insulin levels are higher, and glucose intolerance is apparent, but insulin sensitivity evaluated by ITT is not decreased, compared to that in mice fed NC within 1 week of intervention of the diets [19,20,21,22]. In mice fed HFD for 1 week, β-cell proliferation is increased, with or without an increase of BCM, compared to that in mice fed NC [19,20,21,22]. These results suggest that the increase of BCM by HFD is induced before a decrease of insulin sensitivity. Enhanced expression levels of the genes involved in the cell cycle in islets depends on the term of exposure to HFD. The expression levels of *cyclin D1* and *cyclin D2* mRNA in islets in liver-specific insulin receptor-deficient mice, which show a decrease in insulin sensitivity, are increased, and it has been demonstrated that cyclin D2 contributes to the increase of BCM in these mice [23]. Indeed, the expression levels of *cyclin D1* and *cyclin D2* mRNA in islets of mice fed HFD for 8 or 20 weeks, which show a decrease in insulin sensitivity evaluated by ITT, were higher than those in islets of mice fed NC [18,19]. However, in mice fed HFD for 1 week, which do not display a decrease of insulin sensitivity, the expression levels of *cyclin A2*, but not *cyclin D1* and *cyclin D2*, mRNA in islets are increased [19,20,21,22]. In our previous study, we found that BCM and the number of islets is increased in mice fed HSTD for 22 weeks, which show obesity and hyperglycemia and hyperinsulinemia, and that the expression levels of *cyclin D1* and *cyclin D2* mRNA in islets are increased in these mice [7]. On the other hand, in the present study, we show that BCM and the number of islets is increased in mice fed HSTD for 5 weeks, in which insulin sensitivity evaluated by ITT is not decreased, and that the expression levels of *cyclin A2* mRNA, but not those of *cyclin D1* and *cyclin D2*, mRNA are higher in islets of these mice, compared with those in mice fed NC. These results are similar to the previous experiments in mice exposed to short-term HFD. However, BCM and the number of islets in mice fed HSTD are increased before glucose tolerance and change in body weight appears, unlike those in mice fed NC. These results differ from those in mice exposed to short-term HFD. Plasma insulin levels in HSTD-fed mice under the ad libitum-fed condition were significantly higher compared with those in NC-fed mice, even though insulin sensitivity in peripheral organs, such as skeletal muscle and/or adipose tissue, was increased in HSTD-fed mice after 5 weeks intervention of diet. Whether or not insulin sensitivity is decreased in liver of HSTD-fed mice and/or activation of the hepatic-vagus pathway [24] contributes to the increase of pancreatic BCM and islet number in HSDT-fed mice, should be investigated in future study.

It is reported that switching from HFD to a low-fat diet (LFD) is effective in reducing body weight and improving glucose tolerance in obese mice after long-term exposure to HFD [25,26,27]. In addition, switching from HFD to LFD for 4 weeks normalizes glucose tolerance due to improved GIIS in mice fed HFD for 18 months, although the effect on body weight is comparatively small [25]. The morphology of islets was not been investigated in these reports [25,26,27]. We expected that switching from HSTD to NC in mice fed HSTD for 5 weeks would improve glucose tolerance in the presence of sustained increase in BCM and islet number. However, switching from HSTD to NC for 2 weeks in mice fed HSTD for 5 weeks reversed BCM and islet number to a level equal to that in mice fed NC, and did not alter glucose tolerance. These data suggest that BCM and islet number are regulated by nutrients rather than by a change of body weight or glucose metabolism. The mechanism of reducing BCM and islet number by switching from HSTD to NC for 2 weeks, and the extent of the involvement of nutrients in the regulation of BCM and islet number in humans, should be investigated in a future study. One limitation of the present study is that the effect of increased starch intake was inevitably combined with that of decreased protein intake. Laeger T et al. reported that low-protein diets increase the production and plasma concentration of fibroblast growth factor 21 (FGF21). FGF21 is known to enhance insulin sensitivity through increasing glucose uptake into peripheral tissue [28,29]. Thus, not only a starch-rich diet, but also the reduced protein intake may have contributed to the increased insulin sensitivity found in this study. Another limitation of the present study is that food consumption in mice was not evaluated. Food intake was increased in HSTD-fed mice, compared with that in NC-fed mice 15 weeks after intervention of diet in our previous study [5]. Thus, a difference in food intake may have had some effect on islet morphology. 

A third limitation of the present study is that we did not take account of differences in vitamins, mineral premixes, fiber types, and the amount of fiber in NC and HSTD. Such differences may have affected the microbiome and absorption of nutrients in the intestine [30].

## 5. Conclusions

We investigated the effects of short-term high-starch diet on glucose metabolism and islet morphology. We find that a short-term, high-starch diet increases β-cell mass and islet number concomitantly with increased expression levels of *cyclin A2* and *irs2* mRNA in islets. These changes occurred before changes in blood glucose levels and body weight appeared. Furthermore, switching from the high-starch diet to normal chow for 2 weeks after exposure to this high-starch diet for 5 weeks reversed the β-cell mass and islet number, without altering glucose-induced insulin secretion or body weight.

## Figures and Tables

**Figure 1 nutrients-11-01045-f001:**
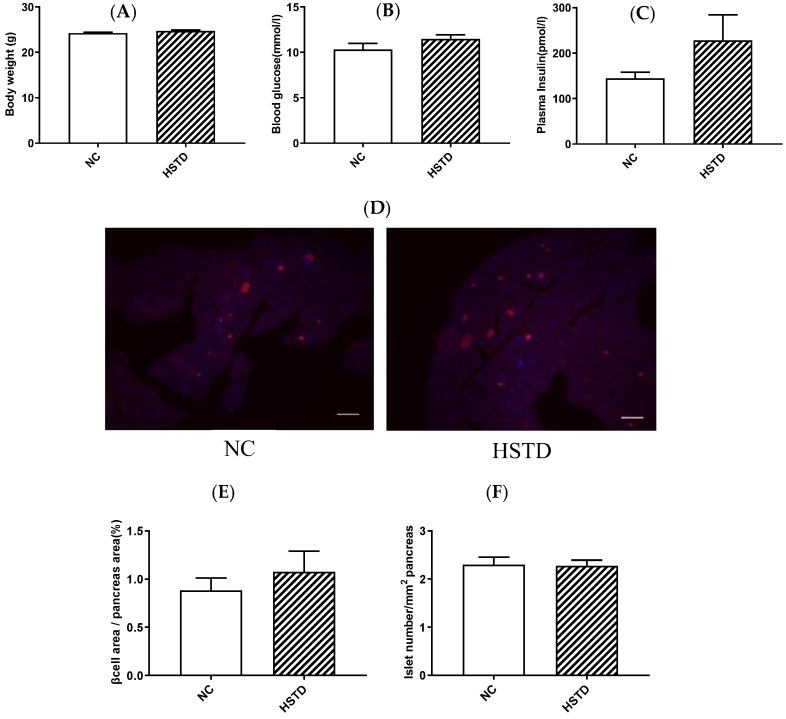
Body weight, metabolic parameters, β-cell mass (BCM), and islet number 3 weeks after intervention of diet in mice. (**A**) Body weight, (**B**) blood glucose and (**C**) plasma insulin levels under ad libitum fed condition. Normal chow (NC)-fed mice (white bar; *n* = 8); high starch, low-protein diet (HSTD)-fed mice (white hatched bar; *n* = 8) (**D**) The pancreatic section was stained with the antibody to insulin (red) and 40,6-diamidino-2-phenylindole (blue). Scale bars, 300 μm. (**E**) β-Cell area relative to pancreas area. (**F**) Number of islets relative to pancreas area. Statistical comparisons were done by unpaired Student’s *t* test in (**A**–**C**,**E**,**F**). Data represent means ± SEM.

**Figure 2 nutrients-11-01045-f002:**
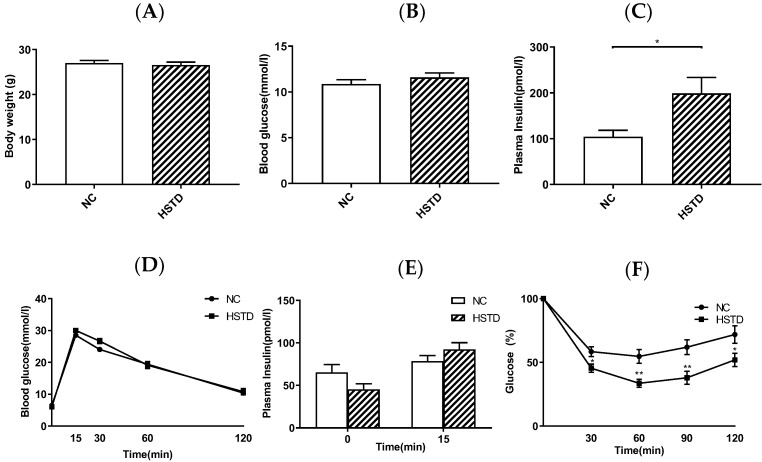
Body weight, metabolic parameters and evaluation of glucose tolerance and insulin sensitivity 5 weeks after intervention of diet in mice. (**A**) Body weight, (**B**) blood glucose and (**C**) plasma insulin levels under ad libitum fed condition. NC-fed mice (white bar; *n* = 8); HSTD-fed mice (white hatched bar; *n* = 7) * *p* < 0.05 (**D**) Blood glucose levels and (**E**) plasma insulin levels during IPGTT in NC-fed mice (black circle and white bar; *n* = 11) and HSTD-fed mice (black square and white hatched bar; *n* = 11). (**F**) Blood glucose levels during ITT in comparison to 0 min in NC-fed mice (black circle; *n* = 12) and HSTD-fed mice (black square; *n* = 9) (* *p* < 0.05, ** *p* < 0.01 compared to NC at indicated time). Statistical comparisons were done by unpaired Student’s *t* test in (**A**–**C**) and multiple *t* tests in (**D**–**F**). Correction for multiple comparisons was performed by using the Holm- Šidák method. Data represent means ± SEM.

**Figure 3 nutrients-11-01045-f003:**
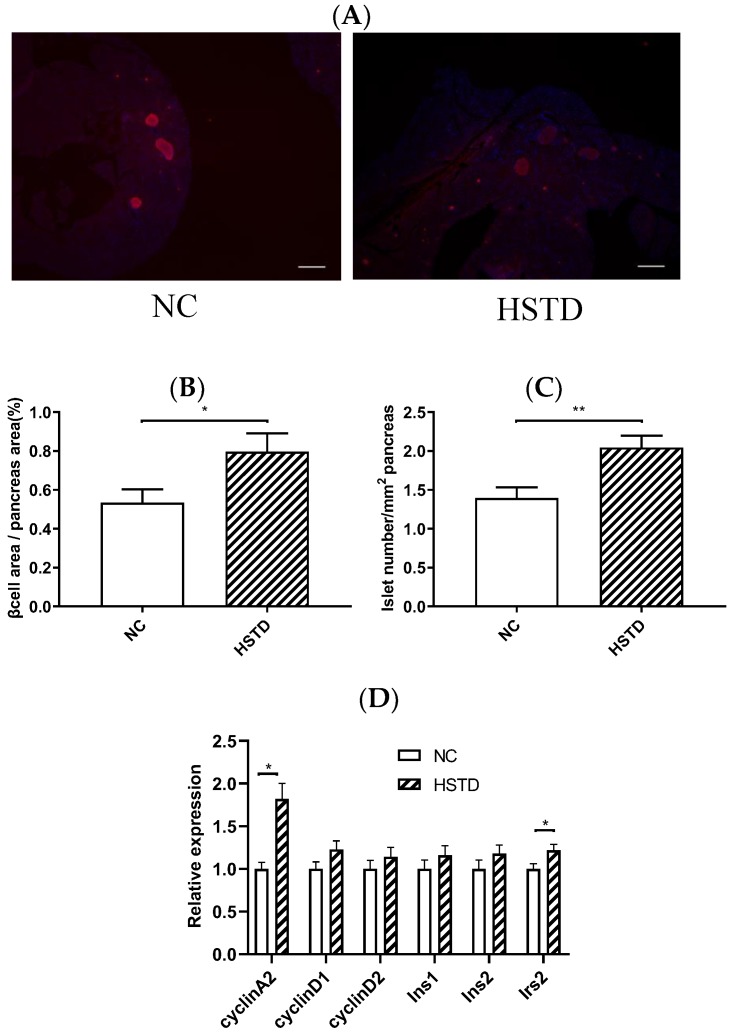
BCM, islet number and mRNA expression levels of islets 5 weeks after intervention of diet in mice. (**A**) The pancreatic section was stained with the antibody to insulin (red) and 40,6-diamidino-2-phenylindole (blue). Scale bars, 300 μm. (**B**) β-Cell area relative to pancreas area. (**C**) Number of islets relative to pancreas area. (**D**) mRNA expression levels of the indicated genes in isolated islets of mice fed NC (white bar; *n* = 16) or HSTD (white hatched bar; *n* = 18) 5 weeks after intervention of diet. Statistical comparisons were done by unpaired Student’s *t* test in (**B**,**C**) and multiple *t* tests in (**D**). Correction for multiple comparisons was performed by using the Holm-Šidák method. Data represent means ± SEM. * *p* < 0.05, ** *p* < 0.01.

**Figure 4 nutrients-11-01045-f004:**
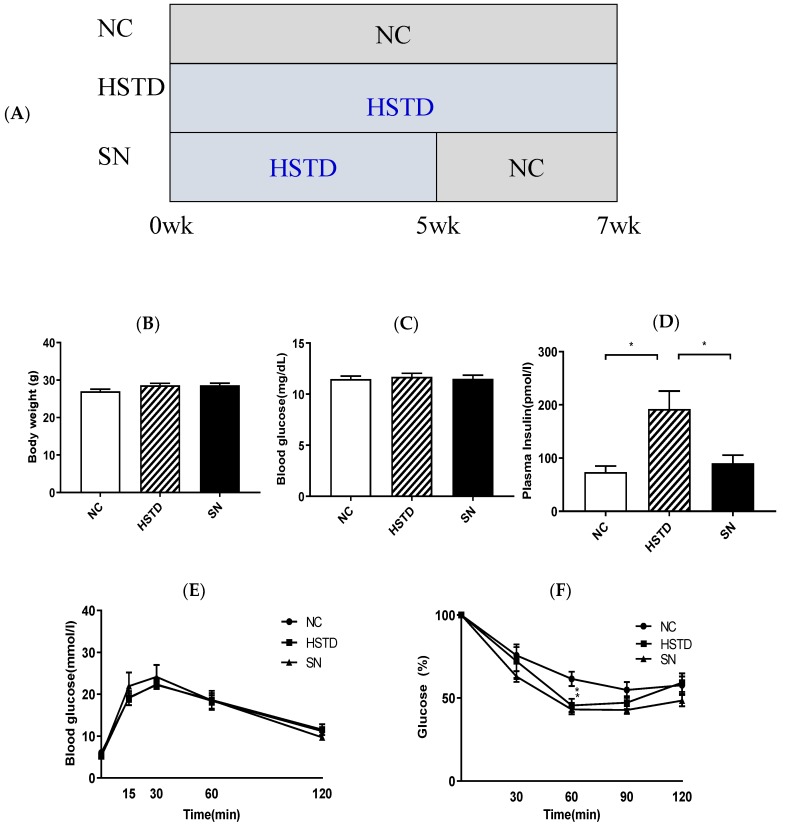
Protocol of this study and the comparison of body weight, metabolic parameters, and morphological analysis of islets between NC group, HSTD group and also mice fed HSTD for 5 weeks followed by NC for 2 weeks (SN) group, 7 weeks after intervention of diet in mice. (**A**) Study design. NC group: Mice fed NC for 7 weeks. HSTD group: Mice fed HSTD for 7 weeks. SN group: Mice fed HSTD for 5 weeks and then fed NC for 2 weeks. (**B**) Body weight, (**C**) blood glucose and (**D**) plasma insulin levels under ad libitum fed condition. NC group (white bar; *n* = 8), HSTD group (white hatched bar; *n* = 8) and SN group (black bar; *n =* 8). (**E**) Blood glucose during IPGTT in NC group (black circle and white bar; *n* = 8), HSTD group (black square and white hatched bar; *n* = 7) and SN group (black triangle and black bar; *n* = 7). (**F**) Blood glucose levels during ITT in comparison to 0 min NC group (black circle; *n* = 8), HSTD group (black square; *n* = 8) and SN group (black triangle; *n* = 8). (* *p* < 0.05 compared to NC at indicated time). (**G**) The pancreatic section was stained with the antibody to insulin (red) and 40,6-diamidino-2-phenylindole (blue). Scale bars, 300 μm. (H) β-Cell area relative to pancreas area and (I) number of islets relative to pancreas area. NC group (white bar; *n* = 12), HSTD group (white hatched bar; *n* = 12) and SN group (black bar; *n =* 12). Statistical comparisons were done by one-way ANOVA, with Tukey’s multiple-comparison test in (**B**–**D**,**H**,**I**), and two-way ANOVA with Tukey’s multiple-comparison test in (**E**,**F**). Data represent means ± SEM. * *p* < 0.05, ** *p* < 0.01, **** *p* < 0.0001.

**Table 1 nutrients-11-01045-t001:** Primers used for quantitative real-time polymerase chain reaction (PCR).

Gene	Forward Primer (5′ to 3′)	Reverse Primer (5′ to 3′)
*βactin*	CATCCGTAAAGACCTCTATGCCAAC	ATGGAGCCACCGATCCACA
*cyclinA2*	TCCTTGCTTTTGACTTGGCT	ATGACTCAGGCCAGCTCTGT
*cyclinD1*	GCCGAGAAGTTGTGCATCTA	TCACCAGAAGCAGTTCCATTT
*cyclinD2*	CACCGACAACTCTGTGAAGC	TCCACTTCAGCTTACCCAACA
*Ins1*	CCTTAGTGACCAGCTATAATCAGAG	AGATGCTGTTTGACAAAAGCC
*Ins2*	AGCCCTAAGTGATCCGCTACA	AAGGTGCTGCTTGACAAAAGCC
*IRS2*	TCCAGGCACTGGAGCTTT	GGCTGGTAGCGCTTCACT

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
