# Peer review of "Short-Term High-Starch, Low-Protein Diet Induces Reversible Increase in β-cell Mass Independent of Body Weight Gain in Mice"

_nutrients, 2019, doi:10.3390/nu11051045_

Round 1
Reviewer 1 Report
Masuda A and colleagues revised their manuscript, which is improved. On the other hand, some concerns still exist.
The authors did not measure food intake in mice treated for 7 weeks with ST. In their former study however, data were collected from mice treated for 15 weeks with ST and these mice showed different food intake compared to NC mice. Therefore it is possible, that the mice consumed different amount of ST diet compare to NC diet during 7 weeks of treatment, which could impact the findings of the current manuscript. Thus, this bias should be discussed as study limitation in the discussion.
The statistical tests are still not clearly indicated in the manuscript. The authors probably used ANOVA with post-hoc test or FDR corrected multiple t-tests for performing several statistical tests in the same time. Statistical tests and especially the applied post-hoc tests should be clearly indicated in figure legends of Fig. 2D, E, F. and Fig. 3D.
In Fig. 2 the letters D, E, F are very close to the figures and should be moved up.
Author Response
Response to Reviewer 1 Comments
Point1: The authors did not measure food intake in mice treated for 7 weeks with ST. In their former study however, data were collected from mice treated for 15 weeks with ST and these mice showed different food intake compared to NC mice. Therefore it is possible, that the mice consumed different amount of ST diet compare to NC diet during 7 weeks of treatment, which could impact the findings of the current manuscript. Thus, this bias should be discussed as study limitation in the discussion.
Response1: We agree with the reviewer that the difference in food consumption may have impacted the findings of the current manuscript. We therefore described this element of bias in the revised discussion section as follows:
Another limitation of the present study is that food consumption in mice was not evaluated. Food intake was increased in HSTD-fed (We changed the abbreviation in accord with reviewer 3’s suggestion) mice compared with that in NC-fed mice 15 weeks after intervention of diet in our previous study (ref. 5). Thus, a difference in food intake may have had some effect on islet morphology.
Point2: The statistical tests are still not clearly indicated in the manuscript. The authors probably used ANOVA with post-hoc test or FDR corrected multiple t-tests for performing several statistical tests in the same time. Statistical tests and especially the applied post-hoc tests should be clearly indicated in figure legends of Fig. 2D, E, F. and Fig. 3D.
Response2: The correction for multiple comparisons was performed by using the Holm-Sidak method in figure legends of Fig. 2D, E, F. and Fig. 3D. We described this in the figure legend.
Point3: In Fig. 2 the letters D, E, F are very close to the figures and should be moved up.
Response3: As the reviewer suggested, we corrected the revised manuscript.
Reviewer 2 Report
The authors considered all suggestions and comments from both reviewers, which helped them to improve the manuscript. At the same time, it now provided enough information on the nutritional interventions and in light of the journal, Nutrients, it is not correct to conclude and state that effects were attributed to the starch content. Rather, as they mention in their responses, it is much more likely due to the reduced protein intake (being below the recommended level for growing mice, so ethically a concern), which alters the focus of the manuscript completely. Taken together, the setup of the study does not allow the authors to make the conclusions as present in the manuscript (title, results, conclusions), and it needs substantial revisions to provide an adequate reflection of the nutritional origin of the effects observed. This is also evidenced by their response to reviewer 2 when discussing ST13 versus ST18: "differential protein content [..] should account for the difference in insulin sensitivity."
The diets are, based on the new information given, not only different in starch and protein amount, but also different in vitamin and mineral levels, as well as in necessary dietary fibers effecting gut microbiota; all simply due to the dilution by adding starch and soy oil to the NC. This is adding to the complicated nutritional interpretation of the studies, which is in its current form way too simplistic.
As a suggestion, switch to a more purified diet composition in order to execute a proper nutritional intervention in mice, as it is more and more acknowledged that chow is not considered an appropriate diet for nutritional studies (see e.g. Pellizzon and Ricci, Nutr Metab 2018; 15:3).
Author Response
Response to Reviewer 2 Comments
Point1: The authors considered all suggestions and comments from both reviewers, which helped them to improve the manuscript. At the same time, it now provided enough information on the nutritional interventions and in light of the journal, Nutrients, it is not correct to conclude and state that effects were attributed to the starch content. Rather, as they mention in their responses, it is much more likely due to the reduced protein intake (being below the recommended level for growing mice, so ethically a concern), which alters the focus of the manuscript completely. Taken together, the setup of the study does not allow the authors to make the conclusions as present in the manuscript (title, results, conclusions), and it needs substantial revisions to provide an adequate reflection of the nutritional origin of the effects observed. This is also evidenced by their response to reviewer 2 when discussing ST13 versus ST18: "differential protein content [..] should account for the difference in insulin sensitivity."
Response1: We agree with the reviewer that reduced protein intake may have affected the findings in the present study such as the difference in insulin sensitivity.
We therefore changed the title to “Short-term high-starch, low-protein diet induces reversible increase in β-cell mass independent of body weight gain in mice.
Point2: The diets are, based on the new information given, not only different in starch and protein amount, but also different in vitamin and mineral levels, as well as in necessary dietary fibers effecting gut microbiota; all simply due to the dilution by adding starch and soy oil to the NC. This is adding to the complicated nutritional interpretation of the studies, which is in its current form way too simplistic.
As a suggestion, switch to a more purified diet composition in order to execute a proper nutritional intervention in mice, as it is more and more acknowledged that chow is not considered an appropriate diet for nutritional studies (see e.g. Pellizzon and Ricci, Nutr Metab 2018; 15:3).
Response2: We agree with the reviewer that it is important to use more purified diets to perform the animal experiments because of the differences in vitamins, mineral premixes, fiber types and the amount of fiber that affect the intestine including gut morphology, microbiome and absorption of nutrients (Pellizzon MA, Ricci MR. Nutr Metab (Lond). 2018;15:3.). We have discussed this shortcoming in the revised discussion section as follows:
A third limitation of the present study is that we did not take account of differences in vitamins, mineral premixes, fiber types and the amount of fiber in NC and HSTD (We changed the abbreviation in accord with reviewer 3’s suggestion). Such differences may have affected the microbiome and absorption of nutrients in the intestine (Pellizzon MA, Ricci MR. Nutr Metab (Lond). 2018;15:3.).
Reviewer 3 Report
Masuda and collaborators provide an interesting study in which metabolic alterations and pancreatic islets alterations induced by high-starch diet (in comparison to a Normal Chow , NC, diet) are studied in mice.
A 3 week high starch regimen is reported not to induce any significant metabolic change or changes in the islet`s structure and number. On the contrary, a prolonged administration of a high-starch diet induces hyperinsulinaemia and at the same time, an improvement in insulin responsiveness, as demonstrated in figure 2F. In parallel, the 5-weeks high starch diet led to an increase in islets number and area and, perhaps as a main finding of this investigation, reversal to a NC diet for two weeks re-established the basal state.
Authors refer to the High Starch Diet as ``ST``. I would suggest changing the abbreviation for this diet as, for example, ``HSD``. In a way, ``ST`` reminds also the idea of ``Standard`` diet and makes the reading of the manuscript uneasy.
Lines 29-281. The authors should discuss and/or propose an hypothesis of why, after 5 weeks of high-starch diet, basal insulinaemia is higher but also peripheral insulin sensitivity is ameliorated. Normally, hyperinsulinaemia is observed when peripheral insulin sensitivity is decreased.
Figure 4F is worrisome. Why there is no increase of the insulin levels 15 minutes after glucose injection? I worry that there is some technical issue here, and the figure 4F panel should be removed.
Overall, the microscopy pictures are of poor quality and they should be better contrasted. Moreover, the pictures in some cases do not seem to mirror the graphical quantifications. It is suggested that the authors choose pictures that represent at best the quantitative data.
While there is some interest in the data reporting on gene expression, the value of these data is somehow of little importance as the differences observed, while statistically significant, appear to be very minor in magnitude, especially for IRS2. Besides, it would be interesting to know whether these gene expression levels return to the basal level after the two – weeks switch to the NC diet.
Other minor comments
The authors should take care to provide a careful proofreading of the manuscript, I point out some examples here
Line 80-81: do the authors mean that 4.5 mg of collagenase are dissolved in 3 ml RPMI? If yes, please remove ``with`` and replace with ``in`` (line 80).
Line 80: place a space between ``4.5`` and ``mg``. Absence of spaces occurs often in the manuscript. Please edit.
Line 85: remove provider reference as already stated in line 84.
Line 96: the primers table, of a small size, could be inserted in the main text, thus avoiding the addition of supplementary material.
Author Response
Response to Reviewer 3 Comments
Point1: A 3 week high starch regimen is reported not to induce any significant metabolic change or changes in the islet`s structure and number. On the contrary, a prolonged administration of a high-starch diet induces hyperinsulinaemia and at the same time, an improvement in insulin responsiveness, as demonstrated in figure 2F. In parallel, the 5-weeks high starch diet led to an increase in islets number and area and, perhaps as a main finding of this investigation, reversal to a NC diet for two weeks re-established the basal state.
Authors refer to the High Starch Diet as ``ST``. I would suggest changing the abbreviation for this diet as, for example, ``HSD``. In a way, ``ST`` reminds also the idea of ``Standard`` diet and makes the reading of the manuscript uneasy.
Response2: We have changed the abbreviation for the high starch diet from ST to HSTD. As HSD is sometimes used as an abbreviation for high-sucrose diet, we used HSTD rather than HSD to avoid confusion.
Point2: Lines 29-281. The authors should discuss and/or propose an hypothesis of why, after 5 weeks of high-starch diet, basal insulinaemia is higher but also peripheral insulin sensitivity is ameliorated. Normally, hyperinsulinaemia is observed when peripheral insulin sensitivity is decreased.
Response2: Plasma insulin levels in HSTD-fed mice under ad libitum-fed condition were significantly higher compared with those in NC-fed mice, even though peripheral insulin sensitivity such as that of skeletal muscle and/or adipose tissue was increased in HSTD-fed mice. Although we cannot perform insulin clamp test in the mice, we believe that insulin sensitivity in liver might be decreased in HSTD-fed mice. We discussed this in the revised discussion section as follows:
Plasma insulin levels in HSTD-fed mice under ad libitum-fed condition were significantly higher compared with those in NC-fed mice, even though insulin sensitivity in peripheral organs such as skeletal muscle and/or adipose tissue was increased in HSTD-fed mice after 5 weeks intervention of diet. Whether or not insulin sensitivity is decreased in liver of HSTD-fed mice and/or activation of the hepatic-vagus pathway (Imai J. Science. 322(5905);1250-4) contributes to the increase of pancreatic BCM and islet number in HSDT-fed mice should be investigated in future study.
Point3: Figure 4F is worrisome. Why there is no increase of the insulin levels 15 minutes after glucose injection? I worry that there is some technical issue here, and the figure 4F panel should be removed.
Response3: We removed Figure 4F from the revised manuscript.
Point4: Overall, the microscopy pictures are of poor quality and they should be better contrasted. Moreover, the pictures in some cases do not seem to mirror the graphical quantifications. It is suggested that the authors choose pictures that represent at best the quantitative data.
Response4: As the reviewer suggested, the microscopy pictures are of insufficient quality. We have added pictures of islets having higher magnification in the revised Figure 4G. We chose pictures demonstrating that the number of islets was increased in HSTD-fed mice 5 or 7 weeks after intervention of diet compared with that in NC or SN-fed mice.
Point5: While there is some interest in the data reporting on gene expression, the value of these data is somehow of little importance as the differences observed, while statistically significant, appear to be very minor in magnitude, especially for IRS2. Besides, it would be interesting to know whether these gene expression levels return to the basal level after the two – weeks switch to the NC diet.
Response5: Although the difference observed in gene expression is relatively small in magnitude, we believe that our findings are of importance. Nevertheless, we agree with the reviewer that it would be interesting to know whether gene expression levels return to basal levels after switching back to NC-diet. Unfortunately, we had no chance to investigate the gene expression levels in the present study.
Other minor comments
The authors should take care to provide a careful proofreading of the manuscript, I point out some examples here
Point6: Line 80-81: do the authors mean that 4.5 mg of collagenase are dissolved in 3 ml RPMI? If yes, please remove ``with`` and replace with ``in`` (line 80).
Response6: As the reviewer suggested, we corrected the revised manuscript.
Point7: Line 80: place a space between ``4.5`` and ``mg``. Absence of spaces occurs often in the manuscript. Please edit.
Response7: As the reviewer suggested, we corrected the revised manuscript.
Point8: Line 85: remove provider reference as already stated in line 84.
Response8: As the reviewer suggested, we corrected the revised manuscript.
Point9: Line 96: the primers table, of a small size, could be inserted in the main text, thus avoiding the addition of supplementary material.
Response9: As the reviewer suggested, we inserted the primers table in the main text.
In addition, we eliminated errors such as the absence of spaces between numbers and units in the revised manuscript.
Round 2
Reviewer 2 Report
The authors modified the title to show the importance of the low protein level in their diets, which they name high starch for obvious reasons. Although this revision now incorporates my concern to some extend, it is fully neglected in the remainder of the body of the manuscript. I recommend to state 'low protein' also upon first introduction in abstract and introduction (first sentence of abstract as example: Long-term exposure to high starch, low protein diet (HSTD)) to correctly introduce the diet used. If not, readers are not correctly introduced to the nature of the diet and by inclusion into the abbreviation HSTD this is then at least acknowledged in a proper manner.
Author Response
Response to Reviewer 2 Comment
Point1: The authors modified the title to show the importance of the low protein level in their diets, which they name high starch for obvious reasons. Although this revision now incorporates my concern to some extend, it is fully neglected in the remainder of the body of the manuscript. I recommend to state 'low protein' also upon first introduction in abstract and introduction (first sentence of abstract as example: Long-term exposure to high starch, low protein diet (HSTD)) to correctly introduce the diet used. If not, readers are not correctly introduced to the nature of the diet and by inclusion into the abbreviation HSTD this is then at least acknowledged in a proper manner.
Response1: As the reviewer suggested, we changed the abbreviation HSTD from high starch diet to high starch, low protein diet in the revised manuscript.

This manuscript is a resubmission of an earlier submission. The following is a list of the peer review reports and author responses from that submission.
Round 1
Reviewer 1 Report
Masuda et al describe a mouse study continuing along their research line focusing on e.g. pancreatic beta cell mass (BCM) and glucose homeostasis. They fed mice for 3, 5, or 7 weeks a high starch diet (called ST) versus a normal chow (NC), with a subgroup of mice switching from ST after 5 weeks to NC for the remainder of 2 weeks. At these 3 timepoints, they report findings on body weight, blood glucose and insulin levels, and pancreatic BCM and islet counts; moreover, GTT and ITT reuslts at 3 weeks and some target gene analysis at week 5 is presented.
However, several flaws exist which need to be addressed in order to adequately be able to judge its scientific merits.
Of major concern is the exact details of the NC and ST diets used; although they reference to 4 (own) papers (of which only reference 5 contains some valuable additional information), none of these papers report detailed information on dietary composition. This is of high importance, as based on the information provided in the current manuscript, they call the ST diet a high starch diet, while simultaneously, this diet is a protein restricted diet. In fact, it is well below the nutritional requirements of mice at this age being 20en% protein, here they provide only 13en%. As such, it is very likely that differences observed are due to protein restriction, not due to a higher starch content in the diet. Moreover, due to the nature of the ingredients, it might also be due to soy polyphenols, which questions whether the choice of the dietary composition is a nutritional adequate choice. This deserves to be at least a major topic in the discussion, and provision of detailed information on diets (w/w %, en%, etc.).
Also of major concern, is the lack of information of the details of the study set-up c.q. design. It appears as a 3 batch study: 1 batch for 3 weeks, one for 5 weeks, and one batch for 7 weeks timepoint. However, then for -as an example given- BW at 3 weeks, data of all batches is available, but not reported. Same at 5 weeks (2 batches), etc. The usage of only subsets of mice for their data presentation becomes evident when looking at Figure 4 : data of n=8 is shown in Figure 4B, C, and D, while this batch of mice is at least n=12 (Figure 4F,G). As it is fully unclear what data is now obtained from which mice, or even from the same mice, it is hard to make correct scientific conclusions.
The last point of major concern is the choice of using beta-actin as reference gene for normalization of RT-PCR data; as the islet sizes increased, it is highly likely, and expected, that also beta-actin will be higher expressed. Thus, it is not a correct reference gene, and usage of this transcript will lead to incorrect conclusions.
Please report the number of days between GTT or ITT and sacrifice of the mice, as this might have an effect on metabolism in the days thereafter. Indeed, following a IPGTT, BW and food intake normally drop and become normalized only after a full week, which seems not to be the timeframe used here based on the restricted information provided.
From a nutritional point of view, linking to point 1 above: what kind of starch is provided to the mice? There are major differences between high versus low digestible starch-containing diets, including microbiota activity, production of short-chain fatty acids, etc. (e.g. Fernandez-Calleja, Sci Rep 2018). Is the starch composition kept identical between the diets used?
Plasma insulin levels are not really informative, as they represent fed state, while fasting levels should be given (Figure 2). Data can be simply explained by a difference in last moment of eating, not at all due to the difference in dietary composition. Indeed, no effects were seen by the IPGTT, which is a highly controlled metabolic flexibility analysis in contrast to the fed ad libitum state. Linked to point 2, are the mice used for Figure 2F the same as those used for 2D? What is total number of animals used for the whole study?
The significant difference in plasma insulin and islet number (4D, 4G) seems more to be caused by a reduction in NC group at 7 weeks than to an increase in ST group when compared to 5 weeks (Figure 2 and 3); this alters the conclusion completely.
Minor point: line 195, please adjust to correct referencing to sub panels. of Figure 4
Reviewer 2 Report
Masuda A and colleagues investigated the role of 7 weeks starch diet (ST) compared with 5 weeks ST and 2 weeks normal chow (NC) reversion and 7 weeks NC feeding of mice and analyzed beta-cell mass, plasma insulin and glucose levels as well as glucose and insulin tolerance. The manuscript is well written, however several concerns are raised.
Major points
The authors compare the effect of ST with NC diet, but the energy density (in kcal/g) is not indicated for the diets. Were the ST and NC diets isocaloric? Did the mice consume similar amount of food, was the energy intake in both groups the same? If the ST and NC diets were not isocaloric or the food consumption was different between the groups, then the reported results on the effect of ST diet could be biased. Furthermore, the reason for using starch diet to study obesity and diabetes it is not clear. Most rodent studies use high-fat diet, which resembles well the “unhealthy” human diet. However diets containing high amount of starch do not belong to the “regular” human diet. Moreover, the major finding of the manuscript that reverting high starch diet is beneficial is not novel, it is somehow trivial.
The data including the elevated insulin level upon ST diet (shown as Figure 2) is not novel, since Sakamoto et al 2012 (PMID: 24843603) also published data on blood glucose, insulin and glucose tolerance upon feeding mice for 5 weeks with ST diet, which was slightly different in protein and fat composition but contained the same amount of starch as carbohydrate source. Is it reasonable to replicate the study of Sakamoto et al? Sakamoto et al also showed that ST diet does not alter insulin tolerance, however the data of the current manuscript suggest that ST diet improves insulin resistance. This discrepancy should be discussed.
The authors showed that ST treatment improved insulin sensitivity after 5 weeks treatment (Fig. 2F), however after 7 weeks no apparent difference is visible in insulin sensitivity (Suppl. Fig. S1C). What is the reason for this discrepancy?
The statistical tests are not clearly indicated in the manuscript. The authors probably used Student´s t-test for comparing two groups but which post-hoc test was applied for comparing three or more groups after ANOVA? To correct for various time points and various treatment conditions for ITT and GTT two-way ANOVA with appropriate post-hoc test should be used. Statistical tests should be indicated in each figure legends.
The materials and methods section is very short and lacks the detailed description of the experiments. “Reference 7” also lacks the important details. Therefore, more details should be given for plasma collection (for insulin measurement), pancreatic islet isolation, real-time PCR. DAPI should be indicated also in the methods section.
In Figure 4E the pictures do not represent that SN treatment decreases islet number and area compared with ST condition, therefore other, representative pictures should be used for this figure.
Minor points
Since Suppl. Fig. 1 denotes important information, it should be moved into the main text.
The description of Fig. 4 in the results section is incorrect, and should be corrected (line 183, 185, 195).
For the pancreatic sections, the heading is missing, and it is not clear whether the pictures are taken from ST or NC animals (Figures 1D, 3A, 4E).
In Fig. 1 and 2 the letters A, B, C, etc are very close to the figures and should be moved up.